# Immunohistochemical analyses reveal FoxP3 expressions in spleen and colorectal cancer in mice treated with AOM/DSS, and their suppression by glycyrrhizin

Guifeng Wang[1], Keiichi Hiramoto[2], Ning Ma[3,4], Shiho Ohnishi[2], Akihiro Morita[2], Yifei Xu[5], Nobuji Yoshikawa[6], Yasuo Chinzei[3], Mariko Murata[5]*, Shosuke Kawanishi[2]*

1 Department of Acupuncture and Moxibustion Medical Science, Suzuka University of Medical Science, Suzuka, Mie, Japan, 2 Faculty of Pharmaceutical Sciences, Suzuka University of Medical Science, Suzuka, Mie, Japan, 3 Graduate School of Health Science, Suzuka University of Medical Science, Suzuka, Mie, Japan, 4 Institute of Traditional Chinese Medicine, Suzuka University of Medical Science, Suzuka, Mie, Japan, 5 Department of Environmental and Molecular Medicine, Mie University Graduate School of Medicine, Tsu, Mie, Japan, 6 Matsusaka R&D Center, Cokey Co., Ltd., Matsusaka, Mie, Japan

* kawanisi@suzuka-u.ac.jp (SK); mmurata@med.mie-u.ac.jp (MM)

## Abstract

We previously demonstrated that glycyrrhizin (GL) suppressed inflammation and carcinogenesis in an azoxymethane (AOM)/dextran sodium sulfate (DSS)-induced murine model of colorectal cancer (CC). In this study, we found an accumulation of regulatory T cells (Tregs) in the spleen and suppression by GL in model mice. ICR mice were divided into four groups: Control, GL, CC, and GL-treated CC (CC+GL), and were sacrificed 20 weeks after AOM/DSS treatment. We measured spleen weight, areas of white and red pulp, and CD8$^+$ T cells (cytotoxic T lymphocytes, CTL), and CD11c-positive cells (dendritic cells) in splenic tissues and forkhead box protein 3 (FoxP3)-positive cells (Tregs) in colorectal and splenic tissues. In all cases, the CC group showed a significant increase compared with those in Control group, and GL administration significantly attenuated this increase. These results indicate that Tregs accumulated in the spleen may participate in inflammation-related carcinogenesis by suppressing CTL. We also suggest that GL which binds to high-mobility group box 1 (HMGB1), suppresses carcinogenesis with decreasing Tregs in the spleen. Furthermore, there was an expression of FoxP3 in cancer cells, indicating that it may be involved in the malignant transformation of cancer cells.

## Introduction

Glycyrrhizin (GL), a triterpenoid glycoside found in licorice root, is 30–50 times sweeter than sugar. GL also has anti-inflammatory, antiviral, anticancer, antiallergic, and hepatocyte regeneration promoting effects [1–7].

The best-known function of GL is its ability to bind to and inhibit high-mobility group box 1 (HMGB1) [8, 9], which is released from the cell nucleus to the extracellular space

**Data Availability Statement:** All relevant data are within the manuscript and its Supporting information files.

**Funding:** This study was partly supported by Japan Society for Promotion of Science under Grant (numbers 19K10585 and 20K20320 to S.K. and 22H03331/23K24589 to M.M.). This study was supported in partly by Cokey Co., Ltd. Under Grant (number 20190401 for N.Y.). The funders had no role in study design, data collection and analysis, decision to publish, or preparation of the manuscript.

**Competing interests:** Author N.Y. was employed by a company Cokey Co., Ltd., operating licorice-related business. N.Y. 's position, etc. does not alter our adherence to PLOS ONE policies on sharing data and materials. Also, there are no restrictions from Cokey Co. Ltd, on all steps of this study. The remaining authors declare that the research was conducted in the absence of any commercial or financial relationships that could be construed as a potential conflict of interest.

during cell necrosis [10–12]. Released HMGB1 induces inflammation [13–17]. The interaction of released HMGB1 with cell surface receptors for advanced glycation end products (RAGE) is one of the major signaling pathways that cause various diseases [9, 18–20]. Binding of HMGB1 to RAGE results in the activation of several signaling molecules, including NF-κB, extracellular signal-regulated kinases (ERK1/2), and p38 [21, 22]. HMGB1 also binds to toll-like receptor (TLR)-2 and TLR-4 and is a potent regulator of tumor necrosis factor (TNF), interleukin (IL)-6, and other proinflammatory cytokines [23, 24]. Therefore, it has been reported that GL binds to HMGB1, thereby inhibiting these signaling pathways and reducing inflammation [25–27]. We have previously demonstrated that GL inhibits this signaling pathway and exhibits cancer suppressive effects in a mouse model of inflammatory colorectal cancer (CC) [28].

Regulatory T cells (Tregs) play an important role in regulating tumor immunity by regulating immune system homeostasis and body immune tolerance [29]. In 2003, Sakaguchi et al. found that the transcription factor, forkhead box protein 3 (FoxP3), acts as a master transcription factor that regulates Treg development, differentiation, and function [30, 31]. FoxP3 belongs to the forkhead transcription factor family and is primarily localized in the nucleus. Recent studies have found FoxP3 expression in invasive Tregs and various cancer cells [32]. Additionally, FoxP3 in cancer cells plays a dual role in malignant transformation as a cancer promoting factor and in suppressing cancer by inhibiting the transcription of downstream oncogenes and related target genes [32].

Our previous study revealed that GL suppressed colitis colon cancer in mice induced with the carcinogen azoxymethane (AOM) and the inflammatory agent dextran sodium sulfate (DSS) [28]. During the course of the study, we found that spleen weight and areas of red and white splenic medulla increased in the spleens of mice with inflammatory CC and that GL suppressed these effects. We investigated the effect of GL on Tregs and CD8$^+$ cells in the spleen as a mechanism for the suppressive effect of GL on inflammatory CC. Furthermore, we examined the immunoreactivity of FoxP3-positive cells in cancer tissues.

## Materials and methods

### Ethics statement

This study complies with the US National Institutes of Health guidelines for the care and use of laboratory animals. Mouse studies were conducted according to procedures approved by the Animal Experimentation Committee of Suzuka University of Medical Science (approval number: 34). Mice were monitored daily by staff at the Suzuka University of Medical Science Animal Experimentation Facility and were killed at the end of the study or at ethical endpoints (signs of deteriorating health, including but not limited to reduced activity, dehydration, abnormal changes in coat, ataxia, and/or excessive weight loss (20% loss of total body weight)). All efforts were made to minimize animal suffering. No deterioration in health was observed in mice during this study. All researchers working with mice received training through the Animal Experimentation Committee of Suzuka University of Medical Science. Animals were killed by overdosing on sodium pentobarbital and tumor tissues and other organs including spleen were harvested. According to the regulations set forth by the Animal Experimentation Committee of Suzuka University of Medical Science, if the tumor grows large and abnormal behavior of the animals is observed, it is recommended to discontinue the experiment from the perspective of animal welfare, in which case the animals are euthanized and considered dead. However, no mice exhibited behavioral abnormalities in this study.

## Sample preparation

In our previous study, we found splenomegaly in mice with AOM/DSS-induced colorectal cancer, and we are interested in this phenomenon and examine the immune cells using formaldehyde-fixed spleens obtained in the study [28]. Briefly, ICR mice (8 weeks old 20 females) were purchased from SLC (Hamamatsu, Japan). The carcinogen AOM was purchased from Sigma Chemical Co. (St. Louis, MO, USA). The inflammatory substance DSS (molecular weight 36,000–50,000) was purchased from MP Biomedicals, Inc. (Solon, OH, USA). GL with 98% purity was purchased from Nagara Science Co., Ltd. (Gifu, Japan). One week prior to the start of the experiment, mice were allowed to acclimatize to the rearing Suzuka University of Medical Science in Shiroko campus animal center environment by freely referencing tap water and pellet food, which was housed in controlled conditions of humidity (50 ± 10%), light (12/ 12 h light/dark cycle), and temperature (22 ± 2°C). ICR mice were randomized according to body weight into four groups (n = 5 per group): Control, GL, CC, and GL-treated CC (CC +GL). Based on the results of the preparatory experiments we minimized the number of samples. we numbered each group of animals and the cages in which they were housed so that researchers were unaware of the grouping from the beginning of the experiment until the time of data analysis.

The CC group (colorectal cancer group) was treated with a single intraperitoneal injection of AOM (10 mg/kg body weight) into the abdominal cavity on the first day, drank distilled water for one week and then received DSS (2%) ad libitum for one week, and after that, they were given distilled water [33]. The CC+GL group was treated with the same as the CC group for up to 2 weeks, and then GL (three times a week; 15 mg/kg/day) was administered orally by sonde for 18 weeks. The Control group was administered saline solution into the abdominal cavity on the first day. The Control group was then given distilled water for 20 weeks. The GL group was given saline intraperitoneally on the first day, followed by distilled water for up to 2 weeks as in the Control group, and then GL was administered orally for 18 weeks. The mice were observed daily throughout the experiment and none were found dead. When animals have reached the end of the study at 20 weeks, the mice were closely monitored for 0.5 to 2 hours after reaching endpoint criteria, and then euthanized by an intraperitoneal overdose of sodium pentobarbital. The colon and other organs including spleen were harvested. We did not measure tibia lengths to normalize spleen weight. However, there was no significant difference of body weight between four groups (Supporting information files). Therefore, it is presumed that spleen weight can be assessed without normalization. Colon and spleen samples were prepared for hematoxylin and eosin (HE) and immunohistochemical (IHC) staining.

## Histopathological and immunohistochemical studies

Tissue samples (n = 3 per group) from the colon and spleen were fixed in 4% formaldehyde, dehydrated and embedded after paraffin wax infiltration before being cut into tissue sections. Colon sections were sectioned to 6 μm and spleen tissue to 3.5 μm thick using Leica RM2265 Microsystems (Wetzlar, Germany). Histopathology of mouse spleens was assessed by HE staining. Subsequent microscopic imaging was performed to determine the area of red pulp (RP) and white pulp (WP) regions in each field of view (2 researchers). The exact procedure of the immunohistochemical staining method in this experiment has been described in the previous article. The primary and secondary antibodies used in this paper were FoxP3 (Abcam, ab75763, 1:300); RAGE (Abcam, ab3611, 1:400); CD8α (Cell signaling, D4W2Z, 1:200); CD11c (cell signaling, 97585, 1:300), avidin-biotin complex (Vectastain ABC kit, Vector Laboratories Burlingame, CA, USA), oxidase DAB kit (Nacalai Tesque Inc.).

For quantitative analysis, all images of the IHC stained sections were obtained with an all-in-one microscope (BZ-X800, KEYENCE, Osaka, Japan) using a Plan Apochromat 20x objective (NA0.75, BZ-PA20, KEYENCE, Osaka, Japan). Two images for each sample were obtained, and the percentage of IHC-staining positive image was automatically quantified by the BZ-X800 Analyzer software (Ver. 1.1.10, KEYENCE, Osaka, Japan).

### Statistical analysis

The results in this experiment were analyzed using SPSS software. We used Student's t-test and the Welch test to analyze the data according to the result of the F-test (equal and non-equal distributions, respectively). Statistical significance was taken as $p < 0.05$. Graphs were also plotted by applying GraphPad Prism 8.

## Results

### Effect of GL administration on the spleen with colon cancer induced by AOM/DSS treatment

Our previous study [28] demonstrated that GL suppress colitis and CC. Fig 1 shows the spleen weights of mice with AOM/DSS-induced CC. The weight of the spleen in the AOM/DSS-treated CC group was significantly higher than that in the Control group ($p < 0.01$), whereas GL markedly attenuated the weight of the spleen in the CC+GL group compared to that in the CC group ($p < 0.01$).

These observations suggest that GL significantly inhibits splenomegaly associated with carcinogenesis in the murine model of colitis-CC.

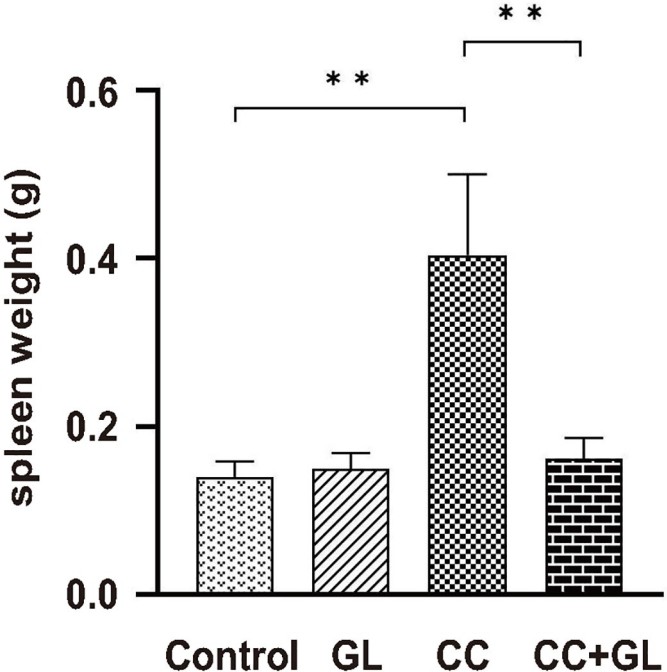

**Fig 1. Effect of GL administration on spleen weight in AOM/DSS-induced colorectal cancer (CC) model mice.**
**$p < 0.01$. AOM, azoxymethane; DSS, dextran sodium sulfate; GL, glycyrrhizin.

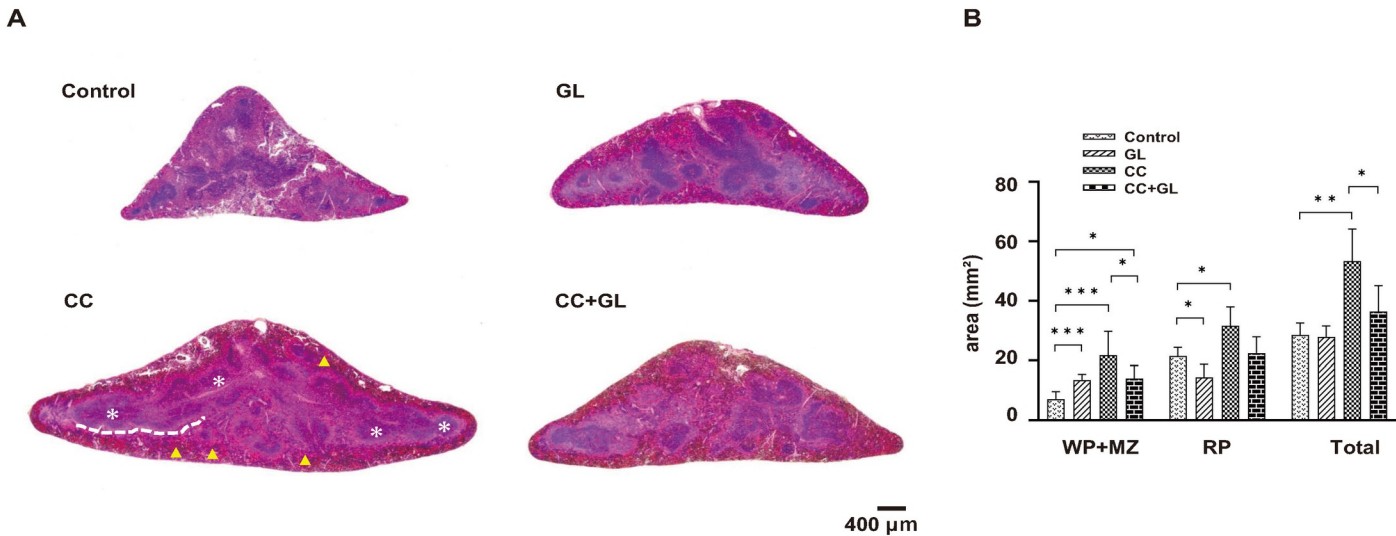

**Fig 2.** Murine splenic tissue with HE staining (A) and its areas (B). White asterisk: white pulp (WP); yellow triangle: red pulp (RP); white dotted line: marginal zone (MZ); Original magnification—20×. Graphs (B) represent the average area (bar: SD; *$p < 0.05$; **$p < 0.01$; ***$p < 0.001$). CC, colorectal cancer; GL, glycyrrhizin; HE, hematoxylin and eosin.

## Histopathological evaluation of the effect of GL administration on the murine spleen

Histological examination of HE stained sections (Fig 2) revealed that the overall area of the splenic tissue (Total; RP and WP with marginal zone (MZ)) increased in the AOM/DSS-induced CC group compared with that in the Control group. GL administration significantly decreased the area of WP ($p < 0.05$) and Total ($p < 0.05$). The area of RP showed a decreasing trend ($p = 0.096$).

These data indicated that GL significantly ameliorated the enlargement of the RP area caused by colitis-associated colorectal tumors in mice.

## Effects of GL on the distribution of Tregs in splenic tissue

FoxP3, a marker of Tregs, was observed in the nuclei of Tregs in the RP (Fig 3A) and WP (Fig 3C) of the splenic tissue (brown staining). The percentage of FoxP3-positive staining area (Tregs) in the RP of the splenic tissue of CC group ($p < 0.01$) was significantly higher than that in the Control group, and GL treatment ($p < 0.01$) decreased this percentage of positive area (Fig 3B). Interestingly, staining in the CC+GL group was lower than that in the CC group, suggesting that GL attenuates Treg accumulation in the CC+GL group. In WP (Fig 3D), the positive staining area percentage of FoxP3 was similar to the results of RP, and the positive staining area of FoxP3 in the CC group was higher ($p < 0.05$) than that of the Control group, while the CC+GL group prevented the increase of positive staining area ($p < 0.05$). This suggests that Treg accumulation is inhibited by GL.

These data indicated that GL treatment resulted in fewer Tregs in the RP of the splenic tissue.

## Effects of GL on the expression of CD8 in splenic tissue

CD8, a marker of cytotoxic T lymphocytes (CTL), was observed in the membranes (brown staining) of the RP (Fig 4A) and WP (Fig 4C).

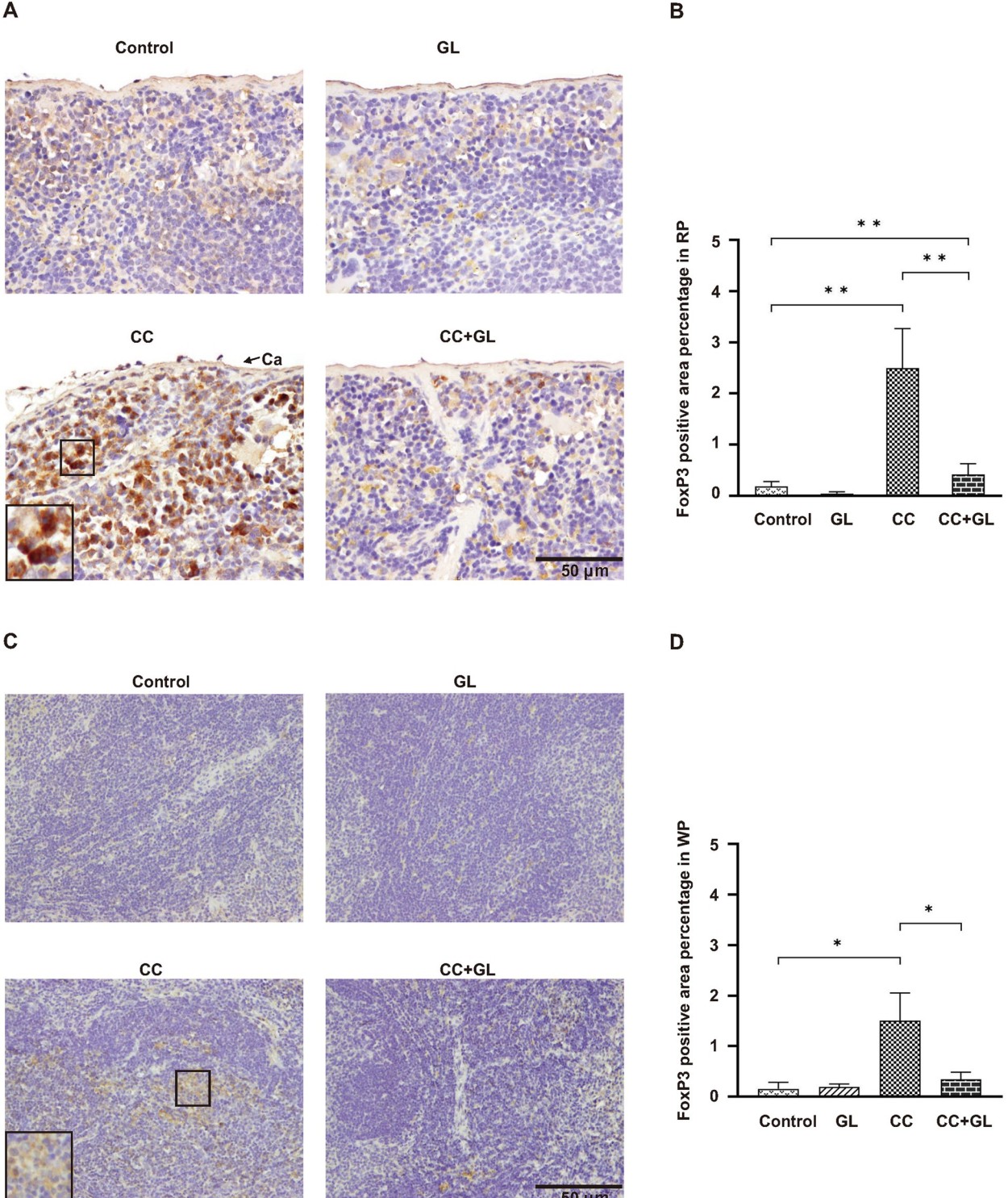

**Fig 3. Immunohistochemical analysis of FoxP3 expression in the splenic tissues of the four groups of mice.** Staining for FoxP3 (A, C) and its percentage of FoxP3-positive area (Tregs) (B, D) in the splenic tissues: RP (A, B) and WP (C, D). Brown color indicates specific immunostaining. Arrow indicates splenic capsule (Ca); Original magnification—200×. Graphs (B, D) represent the average percentage of positive area (bar: SD; $^{**}p < 0.01$; $^*p < 0.05$). CC, colorectal cancer; GL, glycyrrhizin.

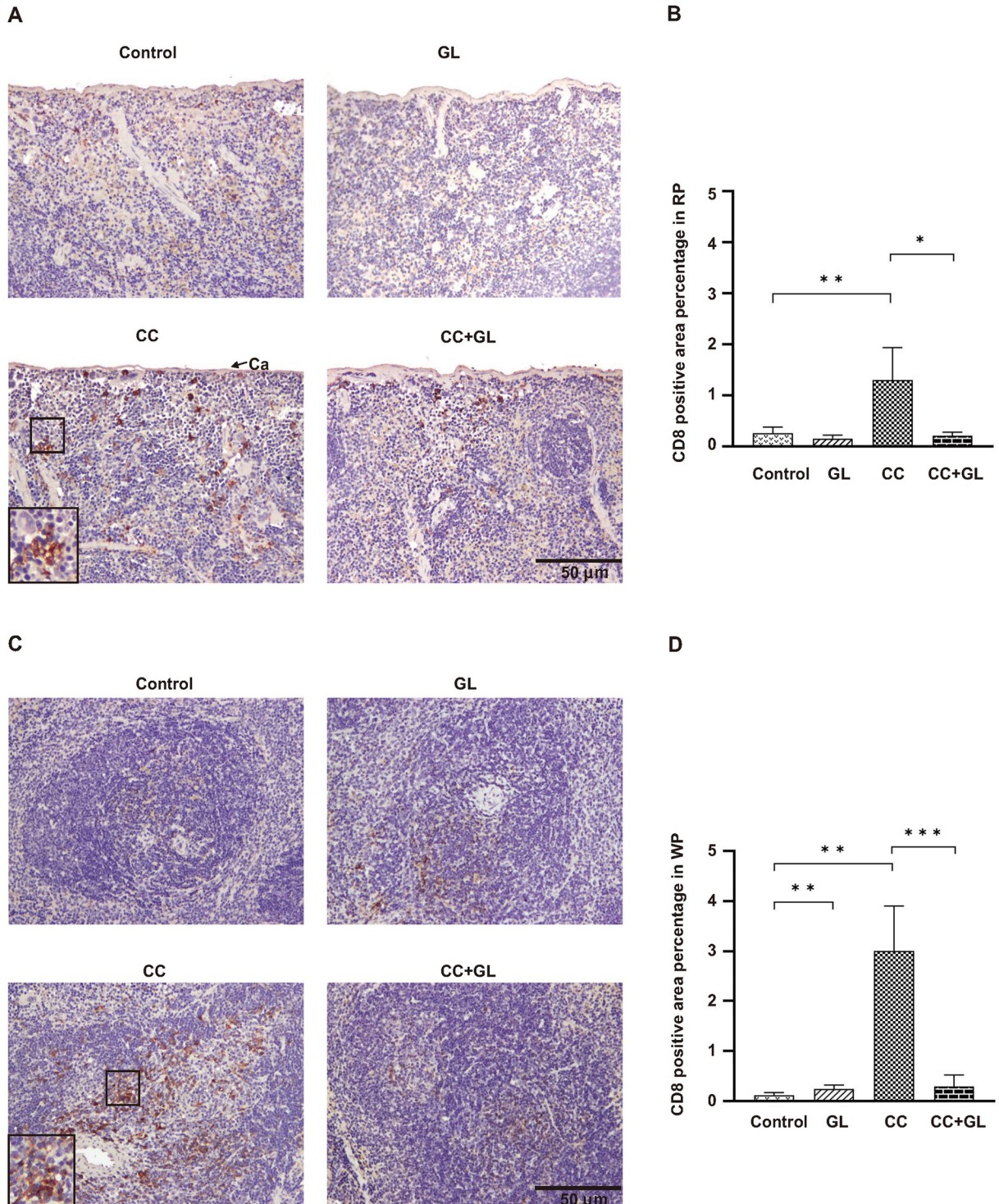

**Fig 4. Immunohistochemical analysis of CD8 expression in the splenic tissues of the four groups of mice.** Staining for CD8 (A, C) and its percentage of positive area (B, D) in the splenic tissues: RP (A, B) and WP (C, D). Brown color indicates specific immunostaining. Arrow indicates splenic capsule (Ca); Original magnification—200×. Graphs (B, D) represent the average percentage of positive area (bar: SD; *$p < 0.05$; **$p < 0.01$; ***$p < 0.001$). RP, red pulp; WP, white pulp; CC, colorectal cancer; GL, glycyrrhizin.

In the RP (Fig 4B) and WP (Fig 4D), IHC staining for CD8 positive area in the Control and GL groups showed a weak staining pattern. The percentage of positive area of CD8 was significantly higher in the CC group than that in the Control group. GL lowered the it positive area of CD8 in the CC+GL group compared to that in the CC group.

These data indicate that the GL treatment resulted in fewer CTLs in both the RP and WP of the splenic tissue.

## Effects of GL on the expression of CD11c in splenic tissue

CD11c, a marker of dendritic cell (DC), was observed in the membranes (brown staining) of cells in the RP (Fig 5A) and WP (Fig 5C). IHC staining of CD11c in the Control and GL groups revealed a weak staining pattern. IHC staining of CD11c in the CC group was higher than that in the Control group, and this increase was reduced in the CC+GL group. The percentage for CD11c positive area in the RP (Fig 5B) and WP (Fig 5D) of the splenic tissue in the CC group were significantly higher than those in the Control group, and this increase was significantly reduced in the CC+GL group. GL lowered the positive area of CD11c in the CC+GL group compared to that in the CC group.

This demonstrated that GL treatment resulted in fewer DCs in both the RP and WP of the splenic tissue.

## Effects of GL on the expression of FoxP3 in colon tissue

IHC staining of FoxP3 (brown staining) in the Control group (Fig 6A) showed little or no staining, and the GL group showed weak staining (Fig 6B). FoxP3 staining was observed in non-cancer tissues surrounding cancer (Fig 6C, arrow) and cancer cells in cancer tissues (Fig 6C and enlarged in Fig 6D). In the CC group, numerous cells showing a stronger immunostaining reaction in the cytoplasm were detected than that in the Control group. In particular, FoxP3 expression was more strongly stained in cancer tissues (Fig 6D) than in the surrounding tissues. However, this increase was prevented in the CC+GL group (Fig 6E).

The percentage for FoxP3 positive area (Fig 6F) in the cancer tissues (CC2) in the CC group was significantly higher than that in the Control group. The percentage of positive area in the CC+GL group were lower than those in the CC group. These data indicate that AOM+DSS treatment may induce the expression of FoxP3 in cells during carcinogenesis and GL that attenuates the expression of FoxP3 in cancer tissues.

## Effects of GL on the expression of RAGE in colon tissue

RAGE, the cell surface receptor for HMGB1, was observed in the membrane and cytoplasm of epithelial cells in colon tissue (brown staining). IHC staining of RAGE in the Control group showed little or no staining (Fig 7A), and the GL group showed weak staining (Fig 7B). In the non-cancer tissue surrounding cancer (Fig 7C, arrow) of the CC group, numerous cells showing a strong immunostaining reaction in the cytoplasm were detected compared with the Control group. In particular, RAGE expression in cancer tissues (Fig 7C and enlarged in Fig 7D) was significantly higher than that in the surrounding cancer tissues. GL lowered the IHC staining of FoxP3 in the CC+GL group (Fig 7E) compared with that in the CC group.

The percentage for RAGE positive area (Fig 7F) in the cancer tissues (CC2) in the CC group were significantly higher than those in the Control group. The percentage of positive area in the CC+GL group were lower than those in the CC group. Relevantly, suppression of RAGE attenuated migration and invasion in breast cancer [34]. These data indicate that AOM +DSS treatment induces RAGE expression in cells during carcinogenesis and that GL attenuates RAGE expression in cancer tissues.

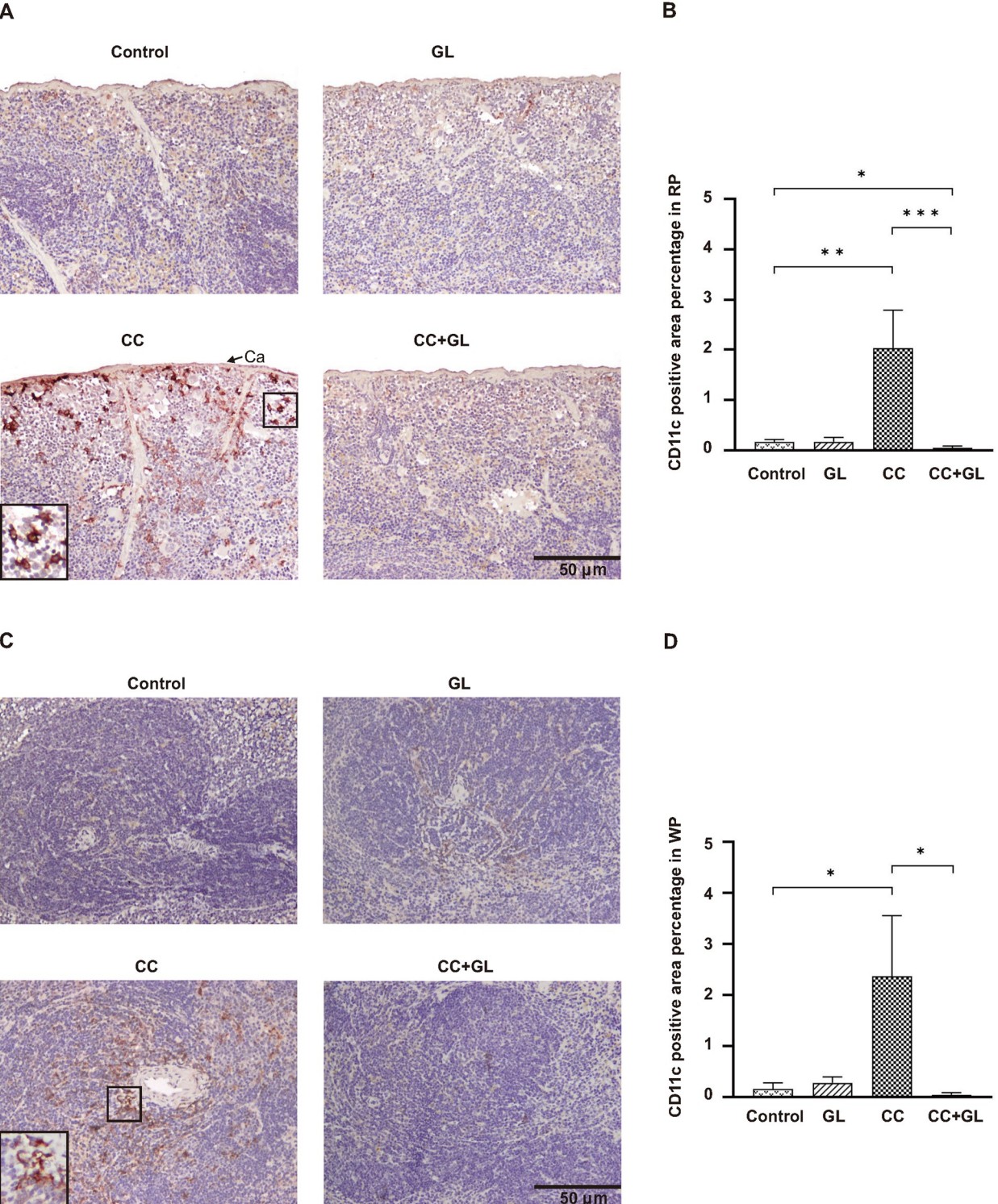

**Fig 5. Immunohistochemical analysis of CD11c expression in the splenic tissues of the four groups of mice.** Staining for CD11c (A, C) and its percentage of positive area (B, D) in the splenic tissues: RP (A, B) and WP (C, D). Brown color indicates specific immunostaining. Arrow indicates splenic capsule (Ca); Original magnification—200×. Graphs (B, D) represent the average percentage of positive area (bar: SD; *$p < 0.05$; **$p < 0.01$; ***$p < 0.001$). RP, red pulp; WP, white pulp; CC, colorectal cancer; GL, glycyrrhizin.

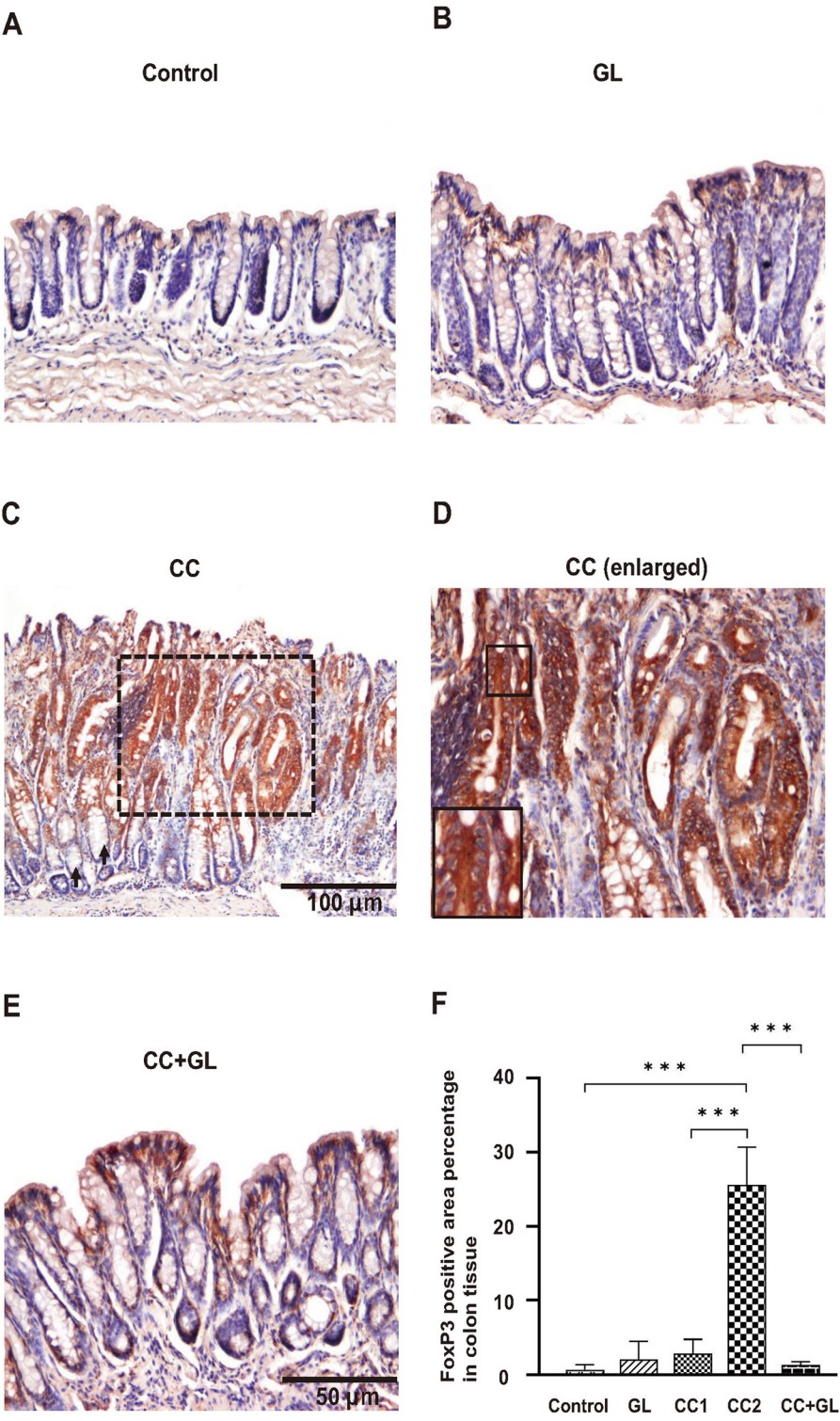

**Fig 6. Immunohistochemical analysis of FoxP3 expression in the colon tissues of the four groups of mice.** Staining for FoxP3 (A-E) and its percentage of positive area (F). Brown color indicates specific immunostaining. Original magnification—200× (A, B, D, E) and 100× (C). (C) Arrows indicate non-cancer tissues (CC1) surrounding cancer area, and dotted square indicates cancer tissue (CC2). (D) Enlarged picture of dotted square in (C). Graphs (F) represent the average percentage of positive area (bar: SD; ***$p < 0.001$). CC, colorectal cancer; GL, glycyrrhizin.

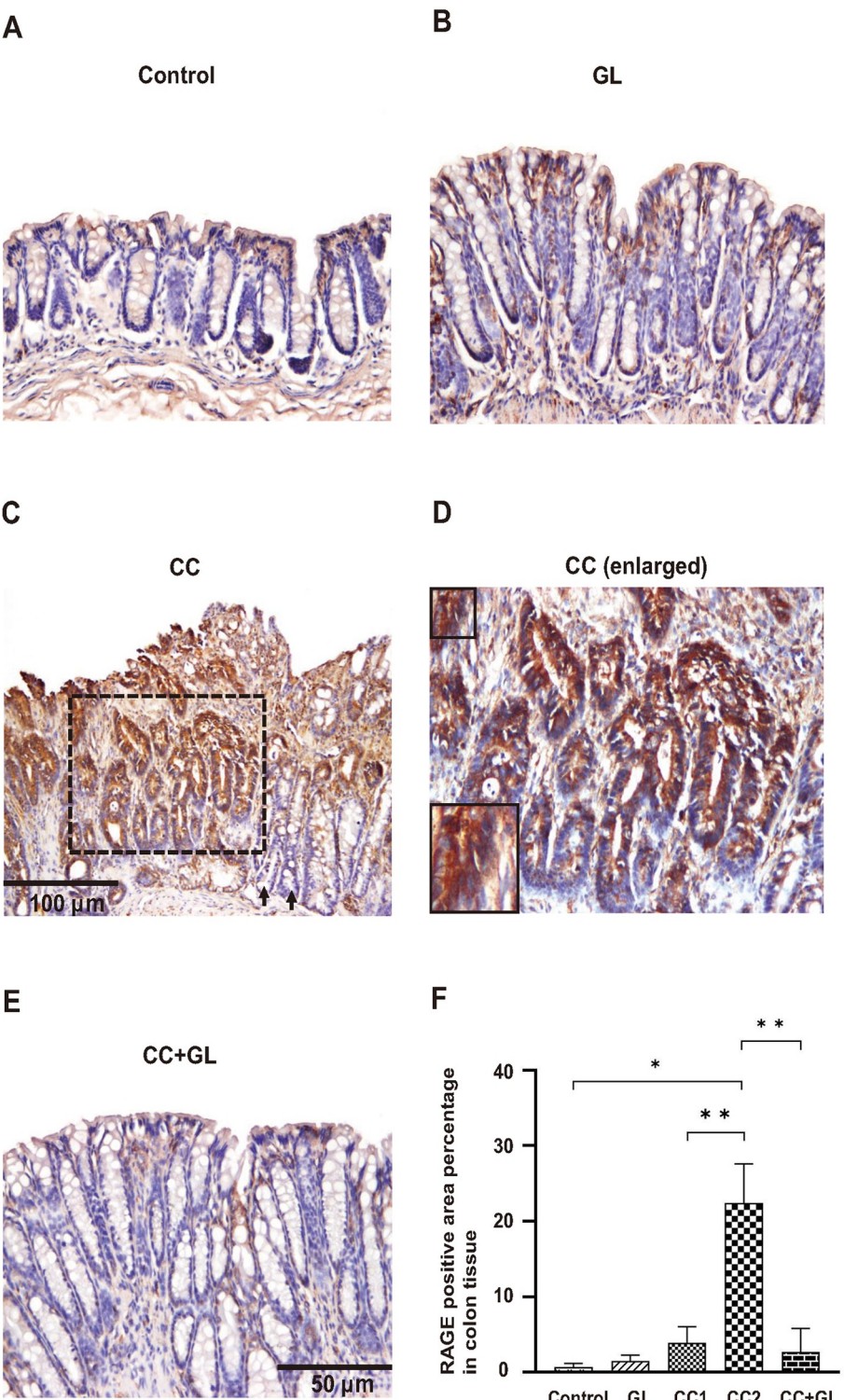

**Fig 7. Immunohistochemical analysis of RAGE expression in the colon tissues of the four groups of mice.** Staining for RAGE (A-E) and its percentage of positive area (F). Brown color indicates specific immunostaining of RAGE. Original magnification—200× (A, B, D, E) and 100× (C). (C) Arrows indicate non-cancer tissues (CC1) surrounding cancer area, and dotted square indicates cancer tissue (CC2). (D) Enlarged picture of dotted square in (C). Graphs (F) represent the average percentage of positive area (bar: SD; *$p < 0.05$; **$p < 0.01$). CC, colorectal cancer; GL, glycyrrhizin.

## Discussion

Our previous study has demonstrated that inflammation-mediated DNA damage is involved of AOM/DSS-induced CC development in addition to AOM-induced DNA damage [28]. As shown in Fig 8, inflammation produces NO and $O_2^-$, causing DNA damage, which in turn leads to gene mutations and cell death as a DNA damage response. The accumulation of genetic mutations leads to carcinogenesis. Increased cell death promotes HMGB1 release. HMGB1 normally resides in the nucleus where it maintains the three-dimensional structure and function of DNA [20, 35]. Once released into the extracellular space, HMGB1 promotes inflammation via TLR2/4 and RAGE, which are expressed in various cells [18], and activated NF-κB pathway results in the release of inflammatory cytokines, such as TNF-α and IL-6, leading to the induction of transcriptional activator STAT3 signaling pathway [36, 37]. Therefore, NO is generated via the IL6-STAT3-iNOS pathway, resulting in DNA damage and malignant transformation of cancer [27, 38, 39]. GL inhibits these signaling pathways by binding to HMGB1 [8]. GL suppresses carcinogenesis in a mouse model of AOM/DSS-induced CC, as previously reported [28].

In this study, we showed that AOM/DSS treatment affected not only colon cancer formation via colitis, but also the spleen, particularly the spleen weight. Furthermore, immunostaining revealed that CD8+ cells (CTLs), CD11c-positive cells (DCs) and FoxP3-positive cells (Tregs), accumulated in the spleen of the CC group in the inflammatory CC model. For efficient antitumor immune responses, CD8+ T cells must be priming and activated toward effector CTLs in a process known as the tumor immunity cycle. CD8+ T-cell priming is essentially directed as a corroboration between cells of innate immunity, including DCs with CD4+ T-cells. Upon activation, effector CTLs infiltrate the core or invading site of the tumor (the so-called infiltrated-inflamed tumor microenvironment) and play an essential role in for killing cancer cells through several mechanisms [40]. Tregs contribute to cancer development and progression by suppressing T effector cell function, thereby compromising tumor killing and promoting tumor growth [41]. FoxP3-expressing Treg cells also effectively suppress tumor

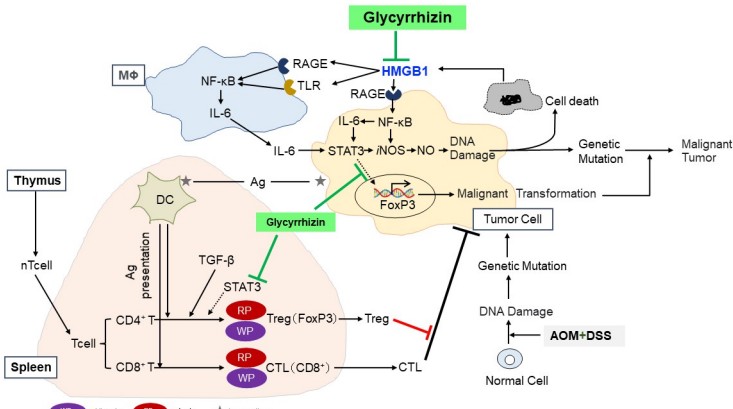

**Fig 8. A possible mechanism of the role of Treg cells and its suppression by GL.** GL has been demonstrated to inhibit a series of HMGB1-TLR4-NF-κB signaling pathways and leaning to the suppression of carcinogenesis via DNA damage. As mechanisms of splenomegaly 1) DCs that have taken up cancer antigens activate CD8+ and CD4+ T cells in the spleen, and the accumulation of CTL and Treg cells in the spleen is the cause of splenomegaly. 2) Treg cells accumulated in the spleen migrate to cancer tissues and promote cancer growth by suppressing immune action by activated CTLs. Ag, antigen; AOM, azoxymethane; CTL, cytotoxic T lymphocytes; DC, dendritic cell; DSS, dextran sodium sulfate; GL, glycyrrhizin; nT, naïve T.

immunity [31]. This suggests that the accumulation of CTL and Tregs is mainly involved in the mechanism of splenic enlargement in inflammatory CC models.

We envision a possible mechanism that in the process of inflammation-related colorectal carcinogenesis via DNA damage through HMGB1-TLR4/RAGE-NF-κB signaling pathways, Tregs, and CTLs, activated by DC-mediated tumor antigen presentation, are accumulated in spleen, and GL inhibits them (Fig 8). The mechanism of splenic enlargement in a mouse model of inflammatory CC is discussed below. Before antigen (Ag) stimulation, naïve T (nT) cells patrol the secondary lymphoid tissues, including the spleen. When DCs take up cancer antigens, they present the antigens and activate T cells that are specific for cancer antigens. Naïve CD8$^+$ T cells and naïve CD4$^+$ T cells are activated to become effector CTL cells and effector Tregs, respectively. In particular, since TGF-β is essential for FoxP3 expression, induction of Treg cell activation occurs in the presence of TGF-β [42, 43]. Activated T cells migrate to the site of the immune response to exert their immune functions. Notably, based on the results of this experiment, it is presumed that DCs that have taken up cancer antigens activate CD8$^+$ and CD4$^+$ T cells in the spleen, and that CTL cells and Treg cells accumulate in the spleen. One of the causes of splenomegaly is the accumulation of T lymphocytes, such as Treg and CTL cells, in the spleen. Under normal conditions, Tregs represent a small T cell population and play key roles in maintaining homeostasis in the immune system. The number of these cells is extensively increased in nearly all cancers, which dampens the response of the immune system against cancer cells [44]. Relatedly, Patwardhan et al. reported that the solid tumor is associated with an increase in the number and function of Tregs [45]. Tregs that accumulate in the spleen may migrate to cancer tissues and promote cancer growth by suppressing the immune actions of CTLs. GL bind to HMGB1 and suppress carcinogenesis by decreasing Tregs in the spleen (Fig 8).

FoxP3 is a master transcription factor of Treg cells that plays an essential role in all aspects of Treg cell differentiation, functional expression, and maintenance of differentiation status. Its expression is thought to be specific to Tregs. Therefore, it can be used as a marker molecule for identifying Tregs. However, FoxP3 has recently been shown to be expressed in various cancer cells [46]. Studies have reported that FoxP3 in cancer cells functions as a tumor suppressor and inhibits the expression of several oncogenes [47]. In contrast, cancer-FoxP3 has been identified as a biomarker associated with poor prognosis in pancreatic cancer, lung cancer, thyroid cancer, and melanoma [48–52]. The inconsistent function of tumor-FoxP3 may be attributable to alternative splicing and posttranslational modifications, that is, FoxP3 with exons 3 and 4 deleted has significantly reduced tumor suppressive ability [53], and phosphorylation of FoxP3 by Lck represses cell invasion [54]. High FoxP3 expression in cancer cells in the CC group and low expression in the CC+GL group, where cancer formation was suppressed by GL, suggest that FoxP3 expression in cancer cells is involved in the malignant transformation of cancer (Fig 8). Additionally, FoxP3 is expressed in the nucleus in these cancers, whereas it is more abundant in the cytoplasm in some cancers such as breast cancer [32, 55] and pancreatic cancer [49]. In this study we found that FoxP3 was expressed in the cytoplasm of CC cells. In many reports, FoxP3 staining in cancer tissue is often interpreted as infiltration of Treg cells [32, 56, 57]. However, in this study, we confirmed FoxP3 expression in the cancer cells themselves, as was evident in the CC group. We also demonstrated RAGE expression in cancer cells, which may indicate that HMGB1-RAGE-NF-κB signaling pathway is activated also in cancer cells. The modulatory effect of STAT3 on FoxP3 expression has been reported on Tregs infiltration in cancer [58, 59]. Notably, Juin et al. [60] demonstrated that GL suppress the expression of pSTAT3 and FoxP3 in melanoma cells in vitro and in Treg cells in vivo. Therefore, HMGB1-RAGE-NF-κB-IL6 pathway may induce FoxP3 expression via STAT3 in murine CC. GL suppressed STAT3-FoxP3 in Tregs and cancer cells, in addition to directly binding to

HMGB1, leading to an anticancer effect (Fig 8). The inhibitory of GL on STAT3 may be supported the report suggesting a role for HMGB1 in the modulation of STAT3 expression [61]. This mechanism is left for future analysis.

The limitations of this study were as follows. To clarify the splenic immune cell population related to colorectal cancer and GL effects, the analysis by flow cytometry might be more informative. However, because of the small sample size, we choose IHC analyses in this study. Therefore, more detailed mechanistic studies are needed to fully understand GL's effects. In addition, colorectal cancer models, such as genetically engineered models and patient-derived xenografts, are the valuable methods to provide a more comprehensive understanding of GL's potential effects. Further studies are needed to determine the clinical relevance of our findings in human colorectal cancer patients.

## Conclusions

Our findings show that spleen enlargement and accumulation of Treg cells and CD8$^+$ cells (CTL) accompany AOM/DSS-induced colon carcinogenesis via HMGB1-TLR4/RAGE-NF-κB signaling pathway. GL inhibit a series of signaling pathways that occur during carcinogenesis by inhibiting HMGB1. Cancer cells themselves expressed FoxP3, which may lead to malignant transformation.

## Supporting information

**S1 File. Original data of the effect of GL administration on spleen weight in AOM/DSS-induced colorectal cancer model mice.** This is the body and spleen weight data.
(XLSX)

**S2 File. Original data of the murine splenic tissue with HE staining and its areas.** This is the spleen area data.
(XLSX)

**S3 File. Original data of the immunohistochemical analysis of FoxP3/CD8/CD11c expression in the splenic tissues and FoxP3/RAGE in colon tissues of the four groups of mice.** This is the FoxP3/CD8/CD11c positive area percentage in spleen (RP/WP) and FoxP3/RAGE positive area percentage in colon data.
(XLSX)

**S4 File. Data analysis of the effect of GL administration on spleen weight in AOM/DSS-induced colorectal cancer model mice.** This is the SPSS output of body/spleen weight data.
(XLSX)

**S5 File. Data analysis of the effect of GL administration on spleen in AOM/DSS-induced colorectal cancer model mice.** This is the SPSS output of spleen area and FoxP3/CD8/CD11c positive area percentage in spleen (RP/WP) data.
(XLSX)

**S6 File. Data analysis of the immunohistochemical analysis of FoxP3/RAGE in colon tissues of the four groups of mice.** This is the SPSS output of FoxP3/RAGE positive area percentage in colon data.
(XLSX)

**S1 Graphical abstract.**
(TIF)

## Author Contributions

**Conceptualization:** Yasuo Chinzei, Mariko Murata, Shosuke Kawanishi.

**Data curation:** Guifeng Wang.

**Formal analysis:** Guifeng Wang.

**Funding acquisition:** Nobuji Yoshikawa, Mariko Murata, Shosuke Kawanishi.

**Investigation:** Guifeng Wang.

**Methodology:** Guifeng Wang, Keiichi Hiramoto, Ning Ma, Shiho Ohnishi, Yifei Xu, Nobuji Yoshikawa.

**Project administration:** Keiichi Hiramoto, Mariko Murata, Shosuke Kawanishi.

**Supervision:** Yasuo Chinzei, Mariko Murata, Shosuke Kawanishi.

**Validation:** Yasuo Chinzei, Mariko Murata, Shosuke Kawanishi.

**Visualization:** Guifeng Wang.

**Writing – original draft:** Guifeng Wang.

**Writing – review & editing:** Shiho Ohnishi, Akihiro Morita, Yasuo Chinzei, Mariko Murata, Shosuke Kawanishi.

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
