## [Decision Letter · Decision Letter 0]

8 May 2024

PONE-D-24-15025Immunohistochemical analyses reveal FoxP3 expressions in spleen and colorectal cancer in mice treated with AOM/DSS, and their suppression by glycyrrhizinPLOS ONE

Dear Dr. Murata,

Thank you for submitting your manuscript to PLOS ONE. After careful consideration, we feel that it has merit but does not fully meet PLOS ONE’s publication criteria as it currently stands. Therefore, we invite you to submit a revised version of the manuscript that addresses the points raised during the review process.

We look forward to receiving your revised manuscript.

Kind regards,

Vinay Kumar, Ph.D.

Academic Editor

PLOS ONE

Journal Requirements:

3. As part of your revision, please complete and submit a copy of the Full ARRIVE 2.0 Guidelines checklist, a document that aims to improve experimental reporting and reproducibility of animal studies for purposes of post-publication data analysis and reproducibility: https://arriveguidelines.org/sites/arrive/files/documents/Author%20Checklist%20-%20Full.pdf Please include your completed checklist as a Supporting Information file. Note that if your paper is accepted for publication, this checklist will be published as part of your article

"This study was partly supported by JSPS KAKENHI under Grant (numbers 19K10585 and 20K20320 to S.K. and 22H03331 to M.M.). This study was supported in partly by Cokey Co., Ltd. Under Grant (number 20190401 for N.Y.)."

6. Thank you for stating the following in the Competing Interests/Financial Disclosure * (delete as necessary) section:

"Conflicts of Interest / COI statement

Author N.Y. was employed by the company Cokey Co., Ltd. The remaining authors declare that the research was conducted in the absence of any commercial or financial relationships that could be construed as a potential conflict of interest."

We note that you received funding from a commercial source: "Cokey Co., Ltd"

7. Your ethics statement should only appear in the Methods section of your manuscript. If your ethics statement is written in any section besides the Methods, please move it to the Methods section and delete it from any other section. Please ensure that your ethics statement is included in your manuscript, as the ethics statement entered into the online submission form will not be published alongside your manuscript.

8. We note that your Data Availability Statement is currently as follows: "All relevant data are within the manuscript and its Supporting Information files."

Reviewers' comments:

Reviewer's Responses to Questions

**Comments to the Author**

1. Is the manuscript technically sound, and do the data support the conclusions?

Reviewer #1: Partly

Reviewer #2: Yes

2. Has the statistical analysis been performed appropriately and rigorously? 

Reviewer #1: Yes

Reviewer #2: Yes

3. Have the authors made all data underlying the findings in their manuscript fully available?

Reviewer #1: Yes

Reviewer #2: Yes

4. Is the manuscript presented in an intelligible fashion and written in standard English?

Reviewer #1: Yes

Reviewer #2: Yes

5. Review Comments to the Author

Reviewer #1: The present manuscript focuses on the effects of glycyrrhizin (GL) on inflammation and carcinogenesis in a murine model of colorectal cancer (CC). Wang and colleagues examine the role of regulatory T cells (Tregs) in this process and the potential mechanisms by which GL exerts its effects. The authors conducted experiments using ICR mice divided into four groups: control, GL alone, CC alone, and CC with GL treatment (CC+GL). After inducing colorectal cancer using azoxymethane (AOM) and dextran sodium sulfate (DSS), they measured various parameters such as spleen weight, white and red pulp areas, as well as different immune cell populations in splenic and colorectal tissues. The study reveals that during AOM/DSS-induced colon carcinogenesis, spleen enlargement and accumulation of Tregs and CD8+ cells (CTL) occur through the HMGB1-TLR4/RAGE NF-κB signaling pathway. GL inhibits these

signaling pathways by targeting HMGB1, potentially impeding carcinogenesis processes.

However, the following issues must be addressed.

1. Did the authors measure the CD4 T cell population in the red pulp? Does GL affect CD4 T effector cells in CC and CC+GL mice? If the authors had used flow cytometry, they might have obtained more precise quantitative data regarding immune cell populations. This represents one of the limitations of the study.

2. GL treatment alone increases the basal levels of CD8 and CD11c in the spleen and Foxp3 and RAGE expression in tumor tissue. Although the authors noted weak staining, could this be due to an overdose or potential side effects of GL? Explain.

3. Is there any difference in the expression of FoxP3 and RAGE in the CC+GL mice group between CC1 and CC2 regions?

4. The method section is not clearly written. How do the authors normalize spleen weights? Do they use tibia lengths?

5. Throughout the manuscript, the authors measured cell counts or semi-quantitative staining grading to assess immune cell populations in IHC staining. They could have quantified the intensity of staining and normalized the total area, potentially avoiding human errors and bias.

6. While the study suggests mechanisms involving Tregs and HMGB1, more detailed mechanistic studies are needed to fully understand GL's effects.

7. Understanding the effects of GL and Tregs in other tissues related to colorectal cancer progression would provide a more comprehensive picture.

8. Using additional colorectal cancer models, such as genetically engineered models or patient-derived xenografts, would provide a more comprehensive understanding of GL's potential effects across different cancer contexts.

9. Further studies are needed to determine the clinical relevance of these findings in human colorectal cancer patients. The authors should mention the limitations of the manuscript.

10. Authors must thoroughly check their manuscript for typos and spacing errors.

Reviewer #2: Authors in this manuscript discuss most possible mechanisms by which Glycyrrhizin might suppress colon cancer development. The study focuses on the role of inflammation and immune response in the process.

The authors made a good colon cancer mice model via Inflammation induced by chemicals (AOM/DSS) damages DNA and promotes colon cancer. Resulting damage triggers a chain reaction involving molecules like NO, HMGB1, and signaling pathways.

Splenomegaly phenotype is seen due to the accumulation of immune cells CTLs, Tregs, DCs responding to inflammation and potential cancer.

Glycyrrhizin appears to disrupt this process by binding to HMGB1 and potentially affecting other signaling pathways, ultimately suppressing Tregs and cancer cell growth.

The data highlights the role of specific molecules HMGB1, NO and immune cells CTLs, Tregs in the process. Mechanism of GL's anticancer effect is proposed with supporting evidence from previous studies and present work.

The exact mechanism by which GL suppresses STAT3 and FoxP3 needs further investigation.

Overall, the manuscript provides a reasonable explanation of a potential mechanism for GL's anticancer properties. It highlights the complex interplay between inflammation, immune response, and cancer development. Further research is needed to validate the proposed mechanism fully. Though not completely new, this study adds a decent piece of knowledge to the existing literature on Glycyrrhizin`s possible mechanism of action.

6. PLOS authors have the option to publish the peer review history of their article (what does this mean?). If published, this will include your full peer review and any attached files.

Reviewer #1: **Yes: **Suman Asalla

Reviewer #2: No

---

## [Author Response · Author response to Decision Letter 0]

14 Jun 2024

Review Comments to the Author

Reviewer #1: The present manuscript focuses on the effects of glycyrrhizin (GL) on inflammation and carcinogenesis in a murine model of colorectal cancer (CC). Wang and colleagues examine the role of regulatory T cells (Tregs) in this process and the potential mechanisms by which GL exerts its effects. The authors conducted experiments using ICR mice divided into four groups: control, GL alone, CC alone, and CC with GL treatment (CC+GL). After inducing colorectal cancer using azoxymethane (AOM) and dextran sodium sulfate (DSS), they measured various parameters such as spleen weight, white and red pulp areas, as well as different immune cell populations in splenic and colorectal tissues. The study reveals that during AOM/DSS-induced colon carcinogenesis, spleen enlargement and accumulation of Tregs and CD8+ cells (CTL) occur through the HMGB1-TLR4/RAGE NF-κB signaling pathway. GL inhibits these signaling pathways by targeting HMGB1, potentially impeding carcinogenesis processes.

However, the following issues must be addressed.

Reply: First of all, we would like to express our deep gratitude for your critical comments and kind advice, especially for comment #5 " They could have quantified the intensity of staining and normalized the total area, potentially avoiding human errors and bias." We quantified again using BZ-X800 Analyzer software (Ver. 1.1.10, KEYENCE, Osaka, Japan) to estimate the IHC staining intensity and normalize the total area as a percentage of the IHC staining positive area. The results by quantitative analyses are similar to the results from semi-quantitative analyses by IHC score. We changed all graphs of IHC score to IHC positive area percentage, to avoid human errors and bias.

1. Did the authors measure the CD4 T cell population in the red pulp? Does GL affect CD4 T effector cells in CC and CC+GL mice? If the authors had used flow cytometry, they might have obtained more precise quantitative data regarding immune cell populations. This represents one of the limitations of the study.

Reply: For this study, we used immunohistochemical analyses because of the limited sample number. Therefore, we could not try flow cytometry for checking the immune cell population including CD4-positive cells. We mentioned about it as the limitations of this study in the discussion part (Lines 384-391). 

2. GL treatment alone increases the basal levels of CD8 and CD11c in the spleen and Foxp3 and RAGE expression in tumor tissue. Although the authors noted weak staining, could this be due to an overdose or potential side effects of GL? Explain.

Reply: Our new quantitative data showed CD8 positive area percentage in WP of GL group was significantly higher than that of the Control group, although there were no significant increases of other markers in GL group. Only by the above data it is difficult to explain it as an overdose or potential side effects of GL.

3. Is there any difference in the expression of FoxP3 and RAGE in the CC+GL mice group between CC1 and CC2 regions?

Reply: CC1 is the non-cancer cells surrounding to cancer cells (CC2). There was no significant difference between CC1 and Control group in the expression of FoxP3 and RAGE. On the other hand, their expressions in CC1 region were significantly lower than that of CC2 region. Therefore, CC1 region in CC group may be comparable to normal colon tissues.

4. The method section is not clearly written. How do the authors normalize spleen weights? Do they use tibia lengths?

Reply: We did not measure tibia lengths, and therefore we could not normalize spleen weight. However, there was no significant difference of body weight between four groups. Since there was no difference in the body weight of the mice, it is presumed that there was a significant difference in spleen weight of CC group.

We mentioned as below: “We did not measure tibia lengths to normalize spleen weight. However, there was no significant difference of body weight between four groups (Supporting Information files). Therefore, it is presumed that spleen weight can be assessed without normalization.” (Lines109–111)

5. Throughout the manuscript, the authors measured cell counts or semi-quantitative staining grading to assess immune cell populations in IHC staining. They could have quantified the intensity of staining and normalized the total area, potentially avoiding human errors and bias.

Reply: Thank you very much your valuable advice. We quantified again using BZ-X800 Analyzer software (Ver. 1.1.10, KEYENCE, Osaka, Japan) to estimate the IHC staining intensity and normalize the total area as a percentage of the IHC staining positive area. We changed all graphs of IHC score to IHC positive area percentage, to avoid human errors and bias.

6. While the study suggests mechanisms involving Tregs and HMGB1, more detailed mechanistic studies are needed to fully understand GL's effects.

Reply: As you pointed, more detailed mechanistic studies are needed to fully understand GL's effects. Therefore, we described it in the discussion part, as the limitations of this study.

7. Understanding the effects of GL and Tregs in other tissues related to colorectal cancer progression would provide a more comprehensive picture.

Reply: To provide a more comprehensive picture to draw the relation of Tregs in other tissues and colorectal cancer and GL effects, many studies, which can elucidate mechanisms, are needed in the future.

8. Using additional colorectal cancer models, such as genetically engineered models or patient-derived xenografts, would provide a more comprehensive understanding of GL's potential effects across different cancer contexts.

Reply: We agree with your comment that colorectal cancer models, such as genetically engineered models or patient-derived xenografts, are the valuable methods to provide a more comprehensive understanding of GL's potential effects.

9. Further studies are needed to determine the clinical relevance of these findings in human colorectal cancer patients. The authors should mention the limitations of the manuscript.

Reply: Thank you for your comment. We described the limitations of this study in the discussion part.

10. Authors must thoroughly check their manuscript for typos and spacing errors.

Reply: We corrected typos and spacing errors throughout the manuscript.

Reviewer #2: Authors in this manuscript discuss most possible mechanisms by which Glycyrrhizin might suppress colon cancer development. The study focuses on the role of inflammation and immune response in the process.

The authors made a good colon cancer mice model via Inflammation induced by chemicals (AOM/DSS) damages DNA and promotes colon cancer. Resulting damage triggers a chain reaction involving molecules like NO, HMGB1, and signaling pathways.

Splenomegaly phenotype is seen due to the accumulation of immune cells CTLs, Tregs, DCs responding to inflammation and potential cancer.

Glycyrrhizin appears to disrupt this process by binding to HMGB1 and potentially affecting other signaling pathways, ultimately suppressing Tregs and cancer cell growth.

The data highlights the role of specific molecules HMGB1, NO and immune cells CTLs, Tregs in the process. Mechanism of GL's anticancer effect is proposed with supporting evidence from previous studies and present work.

The exact mechanism by which GL suppresses STAT3 and FoxP3 needs further investigation.

Overall, the manuscript provides a reasonable explanation of a potential mechanism for GL's anticancer properties. It highlights the complex interplay between inflammation, immune response, and cancer development. Further research is needed to validate the proposed mechanism fully. Though not completely new, this study adds a decent piece of knowledge to the existing literature on Glycyrrhizin`s possible mechanism of action.

Reply: We deeply appreciate your favorable comments. 

6. PLOS authors have the option to publish the peer review history of their article (what does this mean?). If published, this will include your full peer review and any attached files.

Do you want your identity to be public for this peer review? For information about this choice, including consent withdrawal, please see our Privacy Policy.

Reviewer #1: Yes: Suman Asalla

Reviewer #2: No

---

## [Decision Letter · Decision Letter 1]

28 Jun 2024

Immunohistochemical analyses reveal FoxP3 expressions in spleen and colorectal cancer in mice treated with AOM/DSS, and their suppression by glycyrrhizin

PONE-D-24-15025R1

Dear Dr. Murata,

We’re pleased to inform you that your manuscript has been judged scientifically suitable for publication and will be formally accepted for publication once it meets all outstanding technical requirements.

Kind regards,

Vinay Kumar, Ph.D.

Academic Editor

PLOS ONE

Additional Editor Comments (optional):

Reviewers' comments:

Reviewer's Responses to Questions

**Comments to the Author**

1. If the authors have adequately addressed your comments raised in a previous round of review and you feel that this manuscript is now acceptable for publication, you may indicate that here to bypass the “Comments to the Author” section, enter your conflict of interest statement in the “Confidential to Editor” section, and submit your "Accept" recommendation.

Reviewer #1: (No Response)

2. Is the manuscript technically sound, and do the data support the conclusions?

Reviewer #1: (No Response)

3. Has the statistical analysis been performed appropriately and rigorously? 

Reviewer #1: (No Response)

4. Have the authors made all data underlying the findings in their manuscript fully available?

Reviewer #1: (No Response)

5. Is the manuscript presented in an intelligible fashion and written in standard English?

Reviewer #1: (No Response)

6. Review Comments to the Author

Reviewer #1: (No Response)

7. PLOS authors have the option to publish the peer review history of their article (what does this mean?). If published, this will include your full peer review and any attached files.

Reviewer #1: **Yes: **Suman Asalla

---

## [Editor Report · Acceptance letter]

6 Jul 2024

PONE-D-24-15025R1 

PLOS ONE

Dear Dr. Murata, 

I'm pleased to inform you that your manuscript has been deemed suitable for publication in PLOS ONE. Congratulations! Your manuscript is now being handed over to our production team.

Kind regards, 

on behalf of

Dr. Vinay Kumar 

Academic Editor

PLOS ONE